# Personalized Surgery Service in a Tertiary Hospital: A Method to Increase Effectiveness, Precision, Safety and Quality in Maxillofacial Surgery Using Custom-Made 3D Prostheses and Implants

**DOI:** 10.3390/jcm11164791

**Published:** 2022-08-16

**Authors:** Jorge Pamias-Romero, Joan Masnou-Pratdesaba, Manel Sáez-Barba, Alba de-Pablo-García-Cuenca, Sahyly Siurana-Montilva, Anna Sala-Cunill, Victòria Valls-Comamala, Rosa Pujol-Pina, Coro Bescós-Atín

**Affiliations:** 1Oral and Maxillofacial Surgery Service, Hospital Universitari Vall d’Hebron, Vall d’Hebron Barcelona Hospital Campus, Passeig Vall d’Hebron 119-129, 08035 Barcelona, Spain; 2New Technologies and Craniofacial Microsurgery, Vall d’Hebron Institut de Recerca (VHIR), Hospital Universitari Vall d’Hebron, Vall d’Hebron Barcelona Hospital Campus, Passeig Vall d’Hebron 119-129, 08035 Barcelona, Spain; 3Radiology Department, Hospital Universitari Vall d’Hebron, Vall d’Hebron Barcelona Hospital Campus, Passeig Vall d’Hebron 119-129, 08035 Barcelona, Spain; 4MRI Unit, Radiology Department, Hospital Universitari Vall d’Hebron, Vall d’Hebron Barcelona Hospital Campus, Passeig Vall d’Hebron 119-129, 08035 Barcelona, Spain; 5Innovation, Quality and Processes Management, Hospital Universitari Vall d’Hebron, Vall d’Hebron Barcelona Hospital Campus, Passeig Vall d’Hebron 119-129, 08035 Barcelona, Spain

**Keywords:** tertiary hospital, virtual planning, 3D printing, personalized surgery

## Abstract

Personalized surgery (PS) involves virtual planning (VP) and the use of 3D printing technology to design and manufacture custom-made elements to be used during surgery. The widespread use of PS has fostered a paradigm shift in the surgical process. A recent analysis performed in our hospital—along with several studies published in the literature—showed that the extensive use of PS does not preclude the lack of standardization in the process. This means that despite the widely accepted use of this technology, standard individual roles and responsibilities have not been properly defined, and this could hinder the logistics and cost savings in the PS process. The aim of our study was to describe the method followed and the outcomes obtained for the creation of a PS service for the Oral and Maxillofacial Surgery Unit that resolves the current absence of internal structure, allows for the integration of all professionals involved and improves the efficiency and quality of the PS process. We performed a literature search on the implementation of PS techniques in tertiary hospitals and observed a lack of studies on the creation of PS units or services in such hospitals. Therefore, we believe that our work is innovative and has the potential to contribute to the implementation of PS units in other hospitals.

## 1. Introduction

Advances in the field of personalized surgery (PS) involve a “paradigm shift” in the surgical process with respect to conventional surgery techniques. This entails changes in the way the surgical process is planned and performed, which in turn give rise to new workflows that not only involve doctors and surgeons but also engineers and technicians. All these professionals work in joint multidisciplinary teams.

PS involves virtual planning (VP) and the use of 3D printing technology for custom-made elements (known as CAD-CAM [computer-assisted design and computer-assisted manufacturing] technology). PS allows surgeons to develop a virtual surgical plan prior to surgery and to use custom-made surgical devices and surgical prostheses for each patient, with a goal of safer surgeries with more predictable outcomes [1,2,3,4,5,6,7].

PS also makes it possible to achieve complex reconstructions from a structural and geometrical point of view through the design and manufacture of custom-made prostheses and implants that perfectly fit a variety of anatomical defects. Therefore, PS has been widely used in several surgical disciplines, in particular in maxillofacial surgery, mostly for complex reconstructions and in relation to congenital and acquired craniomaxillofacial deformities [8,9,10]. 

In this sense, Lopez et al. describe their use of 3D printing for the treatment of craniomaxillofacial congenital anomalies, including craniosynostosis and microtia. The authors endorse the potential of 3D printing and CAD-CAM techniques for the design of unique scaffolds of any shape or size, offering a personalized approach to patient-specific skeletal defects. They underscore how powerful these techniques are when it comes to the design and reconstruction of complex anatomical sites, such as the ear, in people who suffer from microtia [10].

Day et al. present a series of more than 30 craniofacial defects treated at a tertiary craniofacial referral center using a combination of virtual surgical planning, 3D modelling and patient-specific custom implants. The treated defects were caused either by syndromes (Pierre Robin, Treacher Collins, Apert’s, Pfeiffer, Crouzon) or by other conditions, including craniosynostosis, hemifacial microsomia, micrognathia, multiple facial clefts and trauma. The authors report excellent outcomes for these techniques and mention that complex deformities that require detailed analysis and precise reconstruction benefit the most from the use of advanced 3D techniques. On the basis of the obtained results, the authors conclude that modern 3D technology can potentially improve aesthetic and functional outcomes after complex craniofacial reconstruction, as it allows the surgeon to better analyze complex craniofacial deformities, precisely plan surgical correction with computer simulation of results, customize osteotomies, plan distractions and print custom implants as needed [8].

Other disciplines, such as neurosurgery, traumatology and orthopedic surgery, are increasingly using digital technology and 3D printing to help surgeons minimize human error and reduce surgical time. As published by several authors, the technique is highly reproducible, and it allows for the transfer of the virtually planned steps to the operating table [11,12].

Our hospital has been using PS since 2012, mostly in the Oral and Maxillofacial Surgery Service, and the number of patients treated with PS has increased over time. Although the technique is being used in the hospital on a regular basis, an internal analysis undertaken by the Management Department in the hospital (in collaboration with the professionals who were using PS in 2017) revealed several shortcomings, which are reported below. 

First, a significant shortcoming was the lack of an internal structure to act as an “activity hub” and avoid the dispersion of the different professionals involved in the process. The lack of expert staff (engineers and technicians) to collaborate and provide cross-disciplinary interaction and know-how concentration in the PS process was also a concern. Another limitation was the lack of standardization in the PS process and that of indicators to allow for the proper evaluation and analysis of each step of the process. The abovementioned deficiencies resulted in considerable heterogeneity in the PS surgery process, which in turn caused duplicities in radiological tests, an increase in the time required for the diagnosis and planning of cases and potential delays in surgery scheduling. All these shortcomings were associated with unnecessary costs.

The objective of this article is to describe the PS Service we designed for our hospital, with the aim of correcting the identified drawbacks and undertaking the standardization of the PS process. It is our goal to resolve the lack of internal structure so that all professionals involved are coordinated and all resources and facilities are properly used in order to improve the efficiency and quality of the PS process. 

## 2. Materials and Methods

We created a working group to analyze the shortcomings encountered in the internal analysis and to design a comprehensive PS service to avoid the identified limitations and provide the best PS solution for our hospital. The group published several internal documents regarding the creation of a service from the ground up, how the project would fit into national and European public policies and the suitability of the project for public procurement of innovative process solutions. The project was approved by the Hospital Management Unit and was awarded FEDER funds. Furthermore, a public procurement of innovative process solutions is currently underway [13].

We also wanted to determine and analyze how other tertiary hospitals in our country and in the rest of the world are handling the creation and management of PS services in order to learn from their success stories and from the problems they have encountered. To this end, we performed several literature reviews.

A literature review was performed in the PubMed database regarding the implementation of PS techniques within tertiary hospital centers, including 3D printing and the design of personalized prostheses. The keywords searched initially included personalized surgery, 3D printing and tertiary centers/hospitals, which yielded no results. In order to broaden the search, we added the keywords personalized surgery and 3D printing, which yielded 103 results. These results were then narrowed down by selecting inclusion criteria, including free full-text availability and publication within the last 10 years, leading to a total of 27 papers. Other publications were hand-picked among those articles obtained in different searches (using several keywords closely related to those listed above) performed over a few months. 

Two of the papers were considered relevant to our proposed model of PS. These were an experimental study and a review of 3D-printed surgical implants in a clinical setting and their potential benefits. The authors discuss how 3D printing is commonly used for surgical training and preoperative planning, with very limited clinical applications, and they propose a wider use of these techniques within hospitals and clinics, demonstrating that “manufacturing surgical implants at the clinic with desktop three-dimensional printers can be feasible, effective and economical”. Although this does not exactly reflect our proposed model, it supports the idea that innovative structural and technological advancements within healthcare clinics could be achieved by concentrating many of the involved professionals and procedures internally, leading to a faster and improved service [14,15].

As for the other papers, whereas some pioneering groups have described digital networks in navigation-guided surgery [16,17], we found no articles in which a comprehensive PS service was created from the ground up in a tertiary hospital. Therefore, we decided to perform a new, exhaustive review of the literature by means of a systematic review; our search criteria and strategy are detailed in Appendix A.

We found a total of 109 articles and read the title and abstract of each of them. None of the articles we found provided information on tertiary healthcare centers or hospitals where a PS unit had been created in order to integrate solutions related to PS in several surgical disciplines. Therefore, we consider our work to be innovative and to have the potential to assist other hospitals in their introduction of PS units so that this ever-growing technology can be effectively applied.

## 3. Results

### 3.1. Project Description

The model suggested for the PS service focuses on the creation of a new internal structure in the hospital; a 3D surgical planning and design laboratory (3D-LAB) will be the core of the project and will allow for the integration of all the stages of the PS process at Vall d’Hebron University Hospital (HUVH). The 3D-LAB laboratory will be in permanent contact with the industry (outside the hospital) to exchange information and the customized products that will be used during the PS process (Figure 1).

### 3.2. 3D-LAB Internal Structure and Functional Specifications

The 3D-LAB will have several facilities in place to perform activities undertaken by a multidisciplinary team that will include surgeons, engineers and technicians. 

The facilities are as follows: (1) diagnosis and planning software systems hosted on a central server that allows for direct data import from a corporate storage system (PACS: picture archiving and communication system) in DICOM format (digital imaging and communications in medicine); (2) data export standards (PDF reports and Excel and SPSS spreadsheets); (3) workstations; (4) a 3D printer for prototyping (resin 3D printers) and (5) a data recording system (REDCap database).

The 3D-LAB will be the core and coordinator of the global PS process. Several functions will be performed in the 3D-LAB throughout the process, including interaction with the industry (Figure 2). The functions are described below:Radiological image import from PACS;Processing and merging of images;Diagnosis and planning using diagnosis and planning software;3D prototyping of the required elements for case diagnosis and planning, including resin surgical guides and models that could be required for the placement of CAD-CAM implants or for the use of pre-bent plates;File generation in STL and DICOM formats containing the processed information to be exported to the industry, where the customized elements can be manufactured (using titanium or other materials). At this stage, we will establish online communication with the industry (website connection) to collaborate in the planning and design of the customized elements;Collection of all the customized products that are manufactured outside the hospital (manufacturing outsourced to the industry). All products manufactured by the industry will be sent to the 3D-LAB office for the final stage of surgical treatment, making it possible to follow up on the delivery time for the various products and monitor the case traceability, materials and products throughout the process;Evaluation of results using several established indicators;Establishment of quality control parameters, including the following: delivery time, required regulations and certifications (quality, material biocompatibility and accuracy of measurement), accepted technologies and load/resistance validations; andA communication and networking platform for all professionals involved and establishment of a training plan with respect to PS for all professionals.

The manufacturing stage of customized elements will be outsourced to the industry. Customized elements will be manufactured according to the files created in and sent from the 3D-LAB office, using various manufacturing techniques and materials (mostly titanium or PEEK) according to the required surgery. Manufacturing will take place according to particular quality standards and certifications required for the products ordered.

The final surgical treatment in which the customized elements will be used and implanted will take place in HUVH operating rooms and will be performed by the same professionals who were involved in the PS process.

### 3.3. Evaluation of Result Indicators

Results will be evaluated using several kinds of indicators throughout the stages of the process (stages include diagnosis, planning and design, manufacturing, treatment and post-surgery). These are described in Table 1.

Indicators regarding the quality of service and patient safety will be evaluated all throughout the stages of the PS process. They will include diagnosis and planning time, delivery time for customized products, surgical time, ischemia time (when microsurgical techniques are used), surgical technique reversion, average length of ICU stay and hospital stay, postoperative complications and hospital readmission.

Precision indicators will be used to assess the precision of the surgical technique and that of the manufacturing process of the customized elements, whereas indicators with respect to the effectiveness of the technique will focus on functional evaluation and the quality of life (QoL) of patients; various validated tests and surveys will be completed by patients who undergo surgical interventions involving PS techniques.

Process indicators will be used to monitor compliance during the performance of various stages of the process. Compliance will be monitored using evaluation forms filled-out by the staff who work in the 3D-LAB office.

Technical costs will also be quantified, including the time devoted to the process by professionals, surgical costs, costs of prostheses/implants and other general costs. 

## 4. Discussion

The proposed model is based on the creation of a new in-house structure in the hospital, i.e., a 3D surgical planning and design laboratory (3D-LAB) or office. This laboratory will be the core of the project and will allow for the integration of all the stages of the PS process in our hospital. Specialist surgeons, engineers and technicians will work in the office and collaborate throughout the PS process. Thus, a multidisciplinary team will be created, professionals will work in a hub and knowledge dispersion will be avoided.

The existence of an in-house knowledge hub where cases can be managed among the various specialists involved and where doctors and engineers can closely cooperate may be particularly useful in such a complex field as that of maxillofacial surgery. Even apparently minor aspects of the process may negatively affect its outcomes if approaches from the medical and the engineering fields do not work seamlessly. By way of example, a study by Lo Giudice et al. analyzed the accuracy of a semiautomatic segmentation method in the detection of the volumetric and morphological characteristics of the mandible in comparison with manual segmentation (the gold standard). The study revealed that the area of mismatch between manual segmentation and semiautomatic segmentation was mainly located at the condyle level, with an underestimation of this anatomical region. As stated by the authors, if digital segmentation of the mandible is not accurate, the physical model obtained by 3D printing will not reliably reproduce the anatomy of the mandible, therefore generating discordance between the treatment plan and the clinical outcomes. The authors suggest partnering with companies specialized in 3D imaging technology whenever clinicians need help during the refinement process [20]. In our opinion, the PS service we describe will be very helpful with respect to avoiding having to resort to industry whenever technical issues arise. Some pioneering groups have described digital networks in navigation-guided surgery and the advantages they provide in terms of data exchange, as well as the constant flow of information created by various professionals, which acts as a feedback method for the system [16,17]. 

In this regard, Guijarro-Martínez et al. describe a navigation-assisted multidisciplinary network solution for head and neck cancer that was implemented in their center. According to the authors, the network model stores all the relevant information necessary for each of the involved medical fields in a central server and allows for interactive, multidirectional data flow between all implicated participants [16].

Similarly, Rana et al. describe a language-independent and multidisciplinary imaging-guided navigation technique used in their center for head and neck oncologic surgery. The platform provides intraoperatively collected data to the surgeon, oncologist, radiotherapist, pathologist and radiologist; according to the authors, the platform provides a precise, controlled, safe and minimally invasive surgical method with excellent real-time anatomic orientation [17].

Nevertheless, few studies have been published to date with respect to the introduction of PS units in tertiary healthcare centers or hospitals, as shown by the literature review described above.

Recently, a thought-provoking study fostered by the British Association of Oral and Maxillofacial Surgeons (BAOMS) was published [9,21]. The study examines the barriers to the use of printed titanium and digital planning in maxillofacial surgery in the UK. Results showed that a high percentage of maxillofacial surgeons in the UK (88%) use CAD-CAM technology and design. However, design and manufacturing workflows were found to be highly variable, as were funding opportunities and access to technology. The study highlights the absence of a standardized design pathway for the NHS (National Health Service) in terms of in-house hospital implants, with individual roles and responsibilities. Key barriers include costs, delivery time and the logistical process related to the PS process.

In this sense, we believe that our centralized, comprehensive model for PS could offer significant advantages as compared to current models used to perform PS in most public healthcare tertiary hospitals both in our country and in other countries, such as the UK, as shown in the aforementioned article. In our opinion, these advantages would benefit all stakeholders, including patients, professionals and the hospital.

Patients will benefit from safer, more precise surgeries with improved control of quality indicators and of customized products. The possibility of performing an exhaustive evaluation of the obtained results will also benefit patients.

In our opinion, our PS model could benefit hospital professionals in multiple ways. First, the existence of well-established protocols and circuits managed in a multidisciplinary environment would aid in the professional decision-making process. Decision making would not have to rely so heavily on a single individual but would involve reaching an agreement among different professionals. Secondly, PS would foster learning and training among professionals, which is relevant, considering that new technologies are becoming increasingly relevant and are constantly evolving. Such learning and training would be particularly important in tertiary hospitals that place heavy training burdens on specialists. Thirdly, the potential of establishing a cross-disciplinary collaboration with engineers and specialized technicians could promote concentration of know-how and progress toward research and innovation, in addition to facilitating the creation of new technological developments.

We believe that this model would also provide benefits for the hospital; given that information would be obtained regarding the process, the resources used and the generated costs, the hospital would be provided with an opportunity to achieve improved global management of the PS process. Furthermore, the information obtained with respect to service quality, workflows and time devoted to each stage of the process would allow for the introduction of measures to improve the treatment of various pathologies, including those that are most urgent. From this point of view, the involvement of the hospital in the PS project has made it possible to work with a funding route instead of an individual funding request (IFR), as was done in the past. Whereas when using an IFR inequalities may arise from the free interpretation of what the appropriate route to treat a patient might be, when using a funding route such inequalities are avoided, as the appropriate route to treat a patient is expected to fit into previously established protocols.

As an additional advantage, the new model could provide guidance on the latest regulations regarding new devices and implantable material used in PS, which is an area of concern with respect to quality control. Despite the absence of a standardized in-house implant design pathway with individual roles and responsibilities until recently, recent regulations (ISO 13485) with respect to medical devices (MDR) issued in May 2021 may better bridge the interface between in-house designers and external manufacturers, as the MDR guides the creation of a quality management system for designers and manufacturers of implantable devices [22,23,24,25].

As shown by a study published by Goodson [9], most centers with in-house planning facilities have resin 3D printers (not titanium printers), and they can produce sterilizable resin surgical guides and models that are required for the placement of CAD-CAM implants or for the use of pre-bent plates. In our 3D-LAB laboratory, we will also perform 3D prototyping of several elements considered necessary for case diagnosis and planning, including resin surgical guides and models that could be required for the placement of CAD-CAM implants or for the use of pre-bent plates. For the time being, the printing of customized elements made of titanium or other materials (such as PEEK) will be outsourced to industry. A carefully designed workflow will be followed in our interactions with industry, and we cannot rule out the possibility of printing elements in our laboratory in the future, considering the regulations in place and the potential costs generated.

Finally, we believe that this model would have a positive impact on our healthcare system. The office described above would enable the healthcare system to plan for the provision of PS in all medical specialties, reducing the variability between procedures and allowing for improved control of costs. In turn, these advantages would make it possible to scale up the use of a technology that is steadily on the rise. 

## 5. Conclusions

PS is increasingly used in several surgical specialties, in particular in maxillofacial surgery, where it has achieved the highest level of development. However, some published studies have evidenced the current lack of standardization in the PS process in hospitals, which could have negative repercussions with respect to the logistics and costs involved in the PS process.

The creation of a PS service with an internal structure, a clear definition of functions and the establishment of indicators that allow for the assessment of the global process and for the integration of all professionals involved could offer significant advantages in comparison with the PS models that are currently in place in most tertiary hospitals.

Among the advantages of the new model are improved patient safety and a support system for professionals, both in terms of decision making and as a powerful resource when training others specialists. There would also be advantages for the hospital, such as improved global management of the PS process and additional guidance on the latest regulations with respect to new devices and implantable material used in PS (an area of concern for quality control).

Given the lack of publications (based on our systematic review of the literature) describing the creation of PS units or services in tertiary hospitals, we consider our work to be innovative and to have the potential to contribute to the creation of PS units in other hospitals so that they can introduce this ever-growing technology in their daily work.

## Figures and Tables

**Figure 1 jcm-11-04791-f001:**
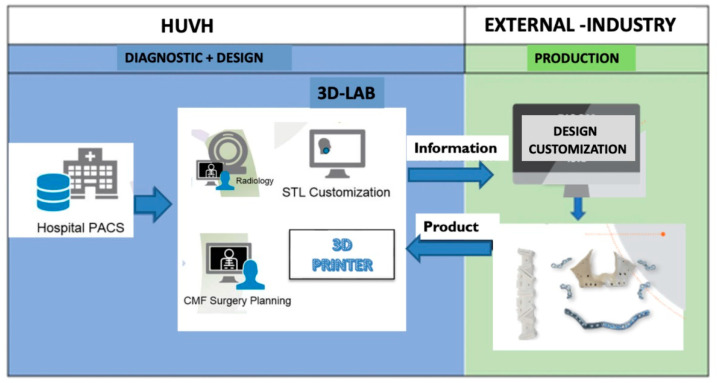
Laboratory of personalized surgery (PS) (3D-LAB).

**Figure 2 jcm-11-04791-f002:**
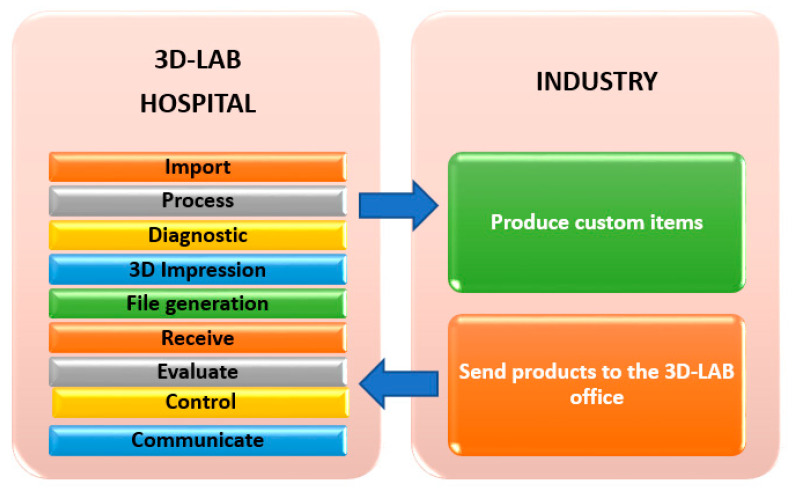
Stages of the personalized surgery (PS) process.

**Table 1 jcm-11-04791-t001:** Indicators for the evaluation of results of personalized surgery (PS).

Indicator Area	Indicator	Definition
Quality of service and patient safety	Customized planning and design time (*Planning and design stage*)	Average time period from the moment the patient is chosen for PS until the design stage is completed
	Delivery time of customized elements (*Manufacturing stage*)	Average manufacturing time from the moment information is sent to the industry until the product is received at the 3D-LAB office
	Surgical time (*Treatment stage*)	Average time from anesthetic induction to end of surgery
	Graft ischemia time (in the event of PS with a microvascularized graft) (*Treatment stage*)	Average time period between the moment the graft is detached from its vessel in the donor site and the moment the anastomosis in the receptor area has been completed and its functionality has been confirmed
	Change of surgical technique (*Treatment stage*)	Percentage of patients in whom we reverted to a conventional surgical technique out of the total of patients who underwent PS
	Average ICU stay (*Treatment stage)*	ICU stay (days) after intervention
	Average hospital stay (*Treatment stage*)	Average hospital stay until discharge after surgery
	Post-surgical complications (*Treatment stage*)	Percentage of patients who suffer complications that arise from PS out of the total number of patients treated with PS [18]
	Hospital readmission (*Treatment stage*)	Percentage of patients who are readmitted to hospital after discharge (48 h post surgery) for reasons related to the surgery out of the total number of discharged patients who underwent PS
Precision	Surgical precision	Degree of precision of the surgical technique (overlapping of pre- and post-surgical images).
	Precision of customized prosthetic elements	Fitting and alignment degree of customized prosthetic elements
Efficacy of the technique	Quality of life (QoL)	Quality of life (QoL) evaluation through tests and surveys [19]
Process	Process indicators	Monitoring of compliance with all stages throughout the process using evaluation forms, as well as monitoring of compliance with the design processes

## Data Availability

Further data may be obtained at https://contractaciopublica.gencat.cat/ecofin_pscp/AppJava/es_ES/notice.pscp?idDoc=71544976&reqCode=viewCtn (accessed on 11 August 2022).

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
