# Peer review of "Personalized Surgery Service in a Tertiary Hospital: A Method to Increase Effectiveness, Precision, Safety and Quality in Maxillofacial Surgery Using Custom-Made 3D Prostheses and Implants"

_jcm, 2022, doi:10.3390/jcm11164791_

Round 1

Reviewer 1 Report

Thank you for the opportunity to review this article.

The study aims to describe the personalized surgery service in the authors' hospital for custom-made 3D prostheses in maxillofacial surgery.

The topic and objective of the paper are very interesting and unconventional. In modern medicine, the introduction of new techniques and technologies is often limited by the complex organziational network required. An article of this nature can certainly support centres that have the potential to approach the topic. Furthermore, the article is in my opinion comprehensive and well written in all its sections. 

I only ask the authors to change the title and abstract to immediately clarify:

1) what is the pertinent field (personalized prostheses in maxillofacial surgery);

2) what is the purpose of the work (to describe a method).

Thank you.

Reviewer 2 Report

Dear authors, 
congratulation for your. The paper is well written and the topic is interesting. Overmore, I suggest you to ampliate the introduction and discussion section. You have also to increase the quality of figure 1, since it is not too clear. In order to give more scientific background to you paper, I suggest you to add the following references, one regarding third molar eruption published in 2003 in Progress, which has the title: "Prognosis of third molar eruption: a comparison of three predictive methods" - M. Manuelli; and the second one regarding the use of 3D in dental practice: "One step before 3D printing-evaluation of imaging software accuracy for 3-dimensional analysis of the mandible: A comparative study using a surface-to-surface matching technique
 (2020) Materials, 13 (12), art. no. 2798". I would like to check again the paper after the authors corrected the manuscript.

Round 2

Reviewer 2 Report

Sem me please the final

pdf approved , Please check the references                                                                                                                              15/16/21  are written correctly 
